# Fruit Structure in Amphicarpic Annual *Gymnarrhena micrantha* (Asteraceae, Gymnarrheneae) in Relation to the Species Biology

Tatyana Kravtsova 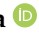

Komarov Botanical Institute, Russian Academy of Sciences, 197022 St. Petersburg, Russia; kraveleon@yandex.ru

**Abstract:** In the amphicarpic annual *Gymnarrhena micrantha* Desf. (Asteraceae), aerial and subterranean fruits differ in morphology, dispersal ability and germination behavior. The aim of our work was to study their structural features in relation to the eco-physiological properties, using light and scanning electron microscopes. Five fruit morphs were found, three of aerial achenes: ebracteate, bracteate and double bracteate ones, and two subterranean fruits with achenes, enveloped in involucral bracts, developed from (I) sessile or (II) not sessile different heads. This species shows divergent fruit differentiation, an increase in their diversity along several lines of morphological differentiation, which corresponds to a multiple seed dispersal and germination strategy. In addition to the already known distinctive features of subterranean achenes (larger size, undeveloped pappus, poor pubescence), they also differ in the simplified structure of the apical and basal achene regions, the absence of the corolla expanded base (cupula) and nectary, other cells parameters in the exotesta and endosperm, another form of the disproportionately developed embryo. The peculiarities of probably subterranean fruit II (seemingly originated through apomixis) extend to various color, pappus structure, sparse pubescence, and the ability of the fruit wall to delaminate. The lack of dense pubescence in the subterranean achenes is a key trait that could lead to increased water permeability of the fruit wall and affect germination rate. Possible adaptive significance of aerial achene structural features is discussed, including specialized corolla cupula, which may be an adaptation to dissemination by rainwater and ants.

**Keywords:** Asteraceae; *Gymnarrhena*; amphicarpy; fruit and seed anatomy; dissemination; germination behavior

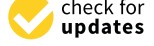



## 1. Introduction

The amphicarpy was defined as formation in plants of both aerial and subterranean fruits [1–4]. It occurs, according to the latest data [4], only in 67 species from 39 genera of flowering plants, belonging mainly to phylogenetically advanced 13 families. There are only two known amphicarpic species in the Asteraceae—*Gymnarrhena micrantha* Desf. (Gymnarrheneae Panero and Funk) [5,6] and *Catananche lutea* L. (Cichorieae Lam. and DC.) [7], in both species it is referred to as amphicarpy sensu lato [8], in which subterranean fruits develop not from subterranean flowers, but rather from chasmogamous ground-level flowers, pulled underground by contractile roots after flowering.

In the Asteraceae, fruit heteromorphism usually follows the form of achenes (cypselae) differences within the capitulum [9,10]. This family surpasses all other families of flowering plants in the number of heterocarpic species, which have revealed a variety of fruit heteromorphism forms [9–14]. Heterocarpy of Asteraceae is associated with heteranthy within the inflorescence and often with the participation of involucral bracts in the formation of marginal achenes; the anatomical differences are more pronounced in fruits that differ greatly in morphology [11].

Heteromorphic fruits and seeds of amphicarpic plants have been understudied anatomically [15–18]. Their ecology and physiology are better examined, including amphicarpic composite species [19–22]. To date, many common features in the reproductive biology

of amphicarpic plants from different families were discovered [3,4,23]. They include the presence of a few, earlier forming and more rapidly germinating subterranean fruits that are larger and heavier than numerous, usually smaller, later-forming and more dormant aerial fruits, as well as the differences between the morphs in the dispersal ability and the ability to form a persistent seed bank (this is typical for aerial fruits). It was found that the seedlings developing from subterranean seeds are less sensitive to germination conditions, larger, more stress-tolerant and competitive than seedlings from aerial seeds. The amphicarpic species were revealed to have a common seed-dispersal dormancy strategy, which differs from the usual strategy in species with dimorphic aerial diaspores. The early production of subterranean fruits and seeds has been called a "pessimistic" strategy in contrast to the optimistic one, i.e., later formation of aerial fruits [24]. It is shown that amphicarpy, like seed heteromorphism in a broad sense [25], is an adaptive strategy for plants living in harsh unpredictable environmental conditions, often in desert regions [26,27]. The amphicarpy was concluded to represent a bet-hedging strategy [4,10,28].

*Gymnarrhena micrantha* is an annual dwarf, usually stemless, ephemeral plant, inhabiting dry regions and deserts from North Africa to Central Asia, the areal of the ancient Mediterranean region [29]. According to the botanical descriptions [1,5,19,20,24,29–38], numerous aerial achenes that develop from pistillate flowers are situated in basal, compact, heterogamous heads (Figure 1A–C). They are hairy, provided with a well-developed pappus and its own subtending bract, which has the shape of a vertically positioned boat, half-surrounding each achene (Figure 2A,B). Staminate flowers, not equipped with bracts, are arranged in loose groups borne on long stalks and concentrated in the center of the head between the achenes. Aerial achenes, produced in spring only in wet years, are dispersed during the winter rainy season by wind and rainwater [20,33–35]; they are also actively collected by ants [39]. Koller and Roth [20] described a series of hygroscopic movements accompanying dissemination in the species, caused by rain and repeated drying, which affects several organs and tissues: receptacle and involucral bracts, receptacular bracts, pericarp hairs and pappus. This phenomenon is similar to hygrochasy (opening of fruits when moistened), since after wetting the fruits are released from the surrounding bract, they acquire an "open" pappus and bent hairs. Freely placed on a convex receptacle, they are ready for dissemination.

Subterranean fruiting heads are few, sessile, and located in the leaf axil on the subterranean part of the stem at a depth of 10–15 mm (Figure 1A,G). In subterranean heads, there are 5–16 flowers, both staminate and pistillate ones [1]; partial subterranean fruiting heads contain 1–3 poorly pubescent achenes (Figure 1H) with an undeveloped pappus, which are enveloped by woody involucral bracts. Their weight (6.50 mg) [20] is significantly greater than the smaller weight of aerial achenes (0.37 mg). Protected by dry woody covers, subterranean achenes remain in the soil under the dead parent plant until germination; due to this reason the plants grow in clusters. The noted differences in the physiology of germination and seedlings development between the aerial and subterranean seeds [20] correspond to the general reproductive patterns of amphicarpic species described above.

There are various hypotheses regarding the appearance of this form of heterocarpy in *G. micrantha* [5]. According to Koller and Roth [20], its biological significance is that two fruit morphs perform two different functions: the aerial fruits are responsible for dispersal of the species and the colonization of new favorable habitats, while the subterranean ones contribute to increasing the probability of the survival of the species. Different strategies of seed dispersal and germination with the presence of amphicarpy are favorable for the successful species survival in desert conditions [26,34,39]. The belowground fruit position "is a very effective method of ensuring atelechory" [2] (p. 94). Zohary [1] believed the greatest biological significance of amphicarpy is that part of the diaspores is kept near the parent plant and is already in the soil; keeping of a favorable area contributes to the survival of the species.

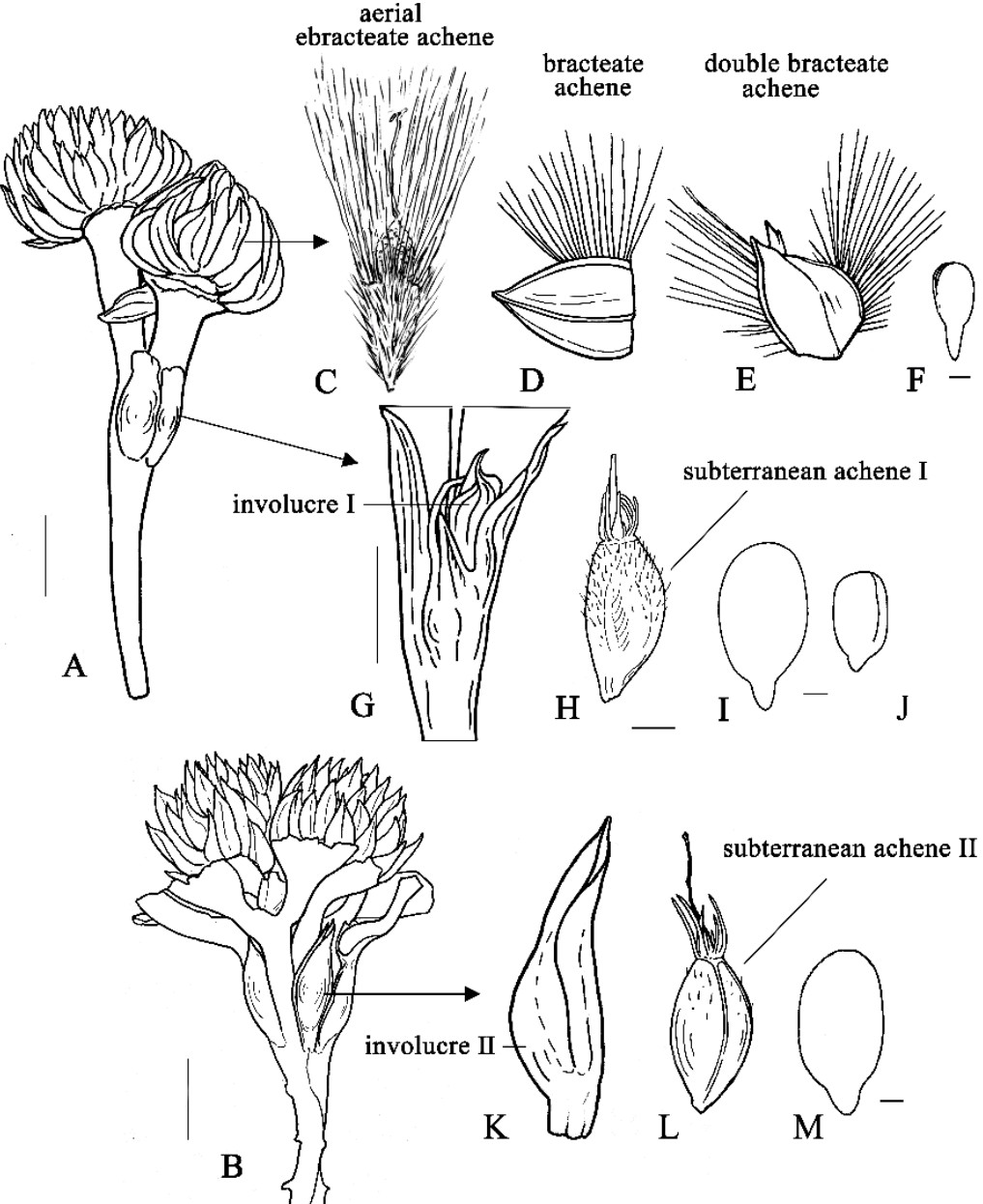

**Figure 1.** Aerial and subterranean fruits and embryos in *Gymnarrhena micrantha*. (**A**) Amphicarpic plant with aerial and subterranean fruiting heads of type I. (**B**) Two plants grown in close proximity of each other and with intertwining stems; one is the same as in Figure 1A, the other with presumably subterranean fruiting heads of type II. (**C–E**) Aerial achenes. (**F**) Embryo from aerial achene. (**G,K**) Subterranean fruiting heads of types I and II, correspondingly. (**H,L**) Subterranean achenes of types I and II, correspondingly. (**I,J**) Embryos from the same partial subterranean head I. (**M**) Embryo from subterranean achene II. Scale bars: 5 mm for (**A,B,G**); 1 mm for (**C–E,H,K,L**); 0.5 mm for (**F,I,J,M**).

*Gymnarrhena* is poorly studied anatomically [5]. Our anatomical and carpological study of the genus *Gymnarrhena* was carried out within the framework of the project "Comparative Anatomy of Seeds" initiated by Academician A.L. Takhtajan, to fill in the gaps in our knowledge about the fruit structure of this genus. The aim of this work was to study the anatomical structure of heteromorphic fruits in *G. micrantha* in relation to dispersal biology and germination behavior.

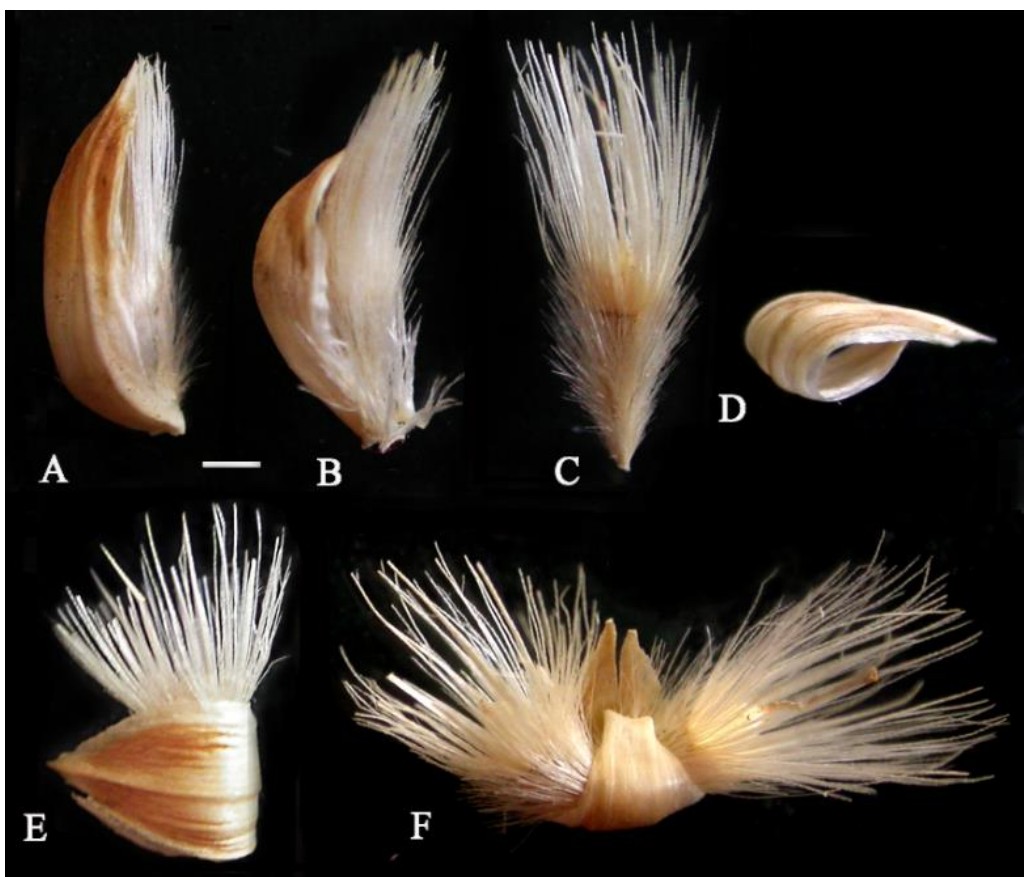

**Figure 2.** Diversity of aerial fruits in *G. micrantha*, light microscopy. (**A**,**B**) Achenes with closed pappus, provided with subtending bract. (**C**) Ebracteate achene with opened pappus. (**D**) Receptacular bract twisted after re-drying. (**E**) Bracteate achene. (**F**) Double bracteate achene. Scale bars = 1 mm.

## 2. Material and Methods

### 2.1. Fruit Material

The study was based on fruits of *G. micrantha* from herbarium specimens kept at the Komarov Botanical Institute LE. Specimens investigated: Egypt, Western Desert, Mediterranean coast, near the village of Burg El Arab, 21.IV.1962, V.P. Botschantzev (LE); B. Balansa. Pl. D'Algérie. 1853. Biskra, 20.II–25.IV (LE); Eastern Persia, Khorasan province, Seistan, on mount Kuh-Khoja, 13.X.1925, E.G. Chernyakovskaya 329 (LE). A total of 2–3 or more aerial achenes of each sample were studied; subterranean fruits were available from one sample only (Egypt, Botschantzev, 1962).

### 2.2. Scanning Electron Microscopy (SEM) and Stereomicroscopy

Fruit and seed morphology and their internal structure on sections were studied with stereomicroscope SteREO Discovery.V20 (Carl Zeiss, Jena, Germany) and scanning electron microscope Jeol JSM-6390 LA. The apical region of aerial achene was also examined using the critical point drying method. For this, the material was fixed with 3% glutaraldehyde solution, washed with phosphate buffer, and then dehydrated in an ethanol, acetone and isoamyl acetate series, critical-point dried in $CO_2$ in Hitachi HCP-2 (Hitachi, Tokyo, Japan), and observed with a scanning electron microscope at 6 kV accelerating voltage. Digital images were prepared using the SEM Control Program associated with the microscope.

### 2.3. Light Microscopy

For anatomical study, fruits were first softened in a mixture of water, 96% ethanol and glycerol in equal proportions. Material was either used fresh or fixed in 70% alcohol and,

for semi-thin sections, in 3% glutaraldehyde and 2% Osmium tetroxide. The longitudinal and transverse sections of fresh material 12 and 24 µm thick were made in the middle part of fruits using freezing microtome, and histochemical studies were carried out to determine lignin (with phloroglucinol and sulphuric acid) and cutin (with Sudan III). Paraffin sections (12 µm thick, stained with Safranin in combination with alcian blue) were made on a rotary microtome following standard procedures [40], and semi-thin sections (2–3 µm thick, stained with toluidine blue) of material embedded in Epon-Araldite epoxy resin were prepared on a Reichert Ultracut R ultramicrotome. Observations were made, and photomicrographs were taken using light microscope AxioImager A1 (Carl Zeiss) equipped with digital imaging AxioCam MRc5 and software Zen 2011. The paper follows the terminology of several authors [11,41–45].

## 3. Results

### 3.1. The Types of Heteromorphic Fruits

In herbarium specimens, the plants of *G. micrantha* (Figure 1A,B) usually have aerial achenes both half-surrounded by the subtending bract, with adpressed pericarp hairs and closed paintbrush pappus (Figure 2A,B), bract-free achenes with an "open" pappus (Figures 1C and 2C), single and "double" bracteate fruits, equipped with a basal appendage from a transversely folded, not detached bract (Figure 2D–F).

Among the subterranean fruits, we distinguish between two types of achenes: (a) subterranean achenes I (Figure 1H), which are poorly pubescent with undeveloped pappus, enveloped in a woody colorless involucre attached to the stem. They are usually present on plants and located in the subterranean fruiting heads, in the leaf axils (Figure 1G). (b) Presumably subterranean achenes II (Figure 1L) that are enclosed in non-sessile bud-like coriaceous dark brown involucre (Figure 1K). They develop from one-flowered partial heads in the branching locations, a few mm below the multi-flowered aerial heads or further down. To clarify the question of their aboveground or underground position, additional observations in nature are required.

### 3.1.1. The Aerial Achenes

Free from subtending bract, hairy pappose aerial achenes (Figures 1C and 2C) are 6–8 mm long, obovate with attenuate base or clavate, round in cross section, 1.0–1.1 mm in diameter, and yellowish. The base of the persistent tubular corolla is transformed into a more or less large cupule with a diameter of 800–1100 µm (Figure 3A–C), varying from small white bulbous (Figure 3C) to large yellowish barrel-shaped in the same plant (Figure 3B), almost equal in length to the achene. The persistent style, enclosed in the corolla tube, protrudes from its narrow distal part and forms two short branches (Figure 3D).

The pappus is white, 4.5–5.5 mm long, heteromorphic (Figure 3A–G), includes 7–8 long lanceolate scales of the inner row and numerous longer bristles of four outer rows attached to the edge of the apical plate. The bristles are thin, irregularly cylindrical, scabrid with teeth all over the surface (Figure 3E). The bristles and scales partially fuse at the base, while each scale is combined with 10 bristles of different diameters. The pappus varies in thickness; in different achenes, the width of the scales in their middle ranges from 280 to 460 µm. Twin hairs of the pericarp are long (0.5–2.0 mm in length), thin, white, with a more or less thickened base (Figure 3H,I), apically shortly forked. The achene base is narrowed into a small yellowish stalk, the carpopodium is not developed. The detachment area is adaxial in the terminology of Dittrich cited in Häffner [43]. The aerial achenes are uniform within the head, the marginal ones are more bent and slightly smaller than the central ones (6–7 mm long). The aerial embryos (Figure 1F), freely located in the achene cavity, are narrow, tooth-shaped, with a long-pointed hypocotyl-root axis, almost equal in length to the cotyledons, and sometimes dissected at the top.

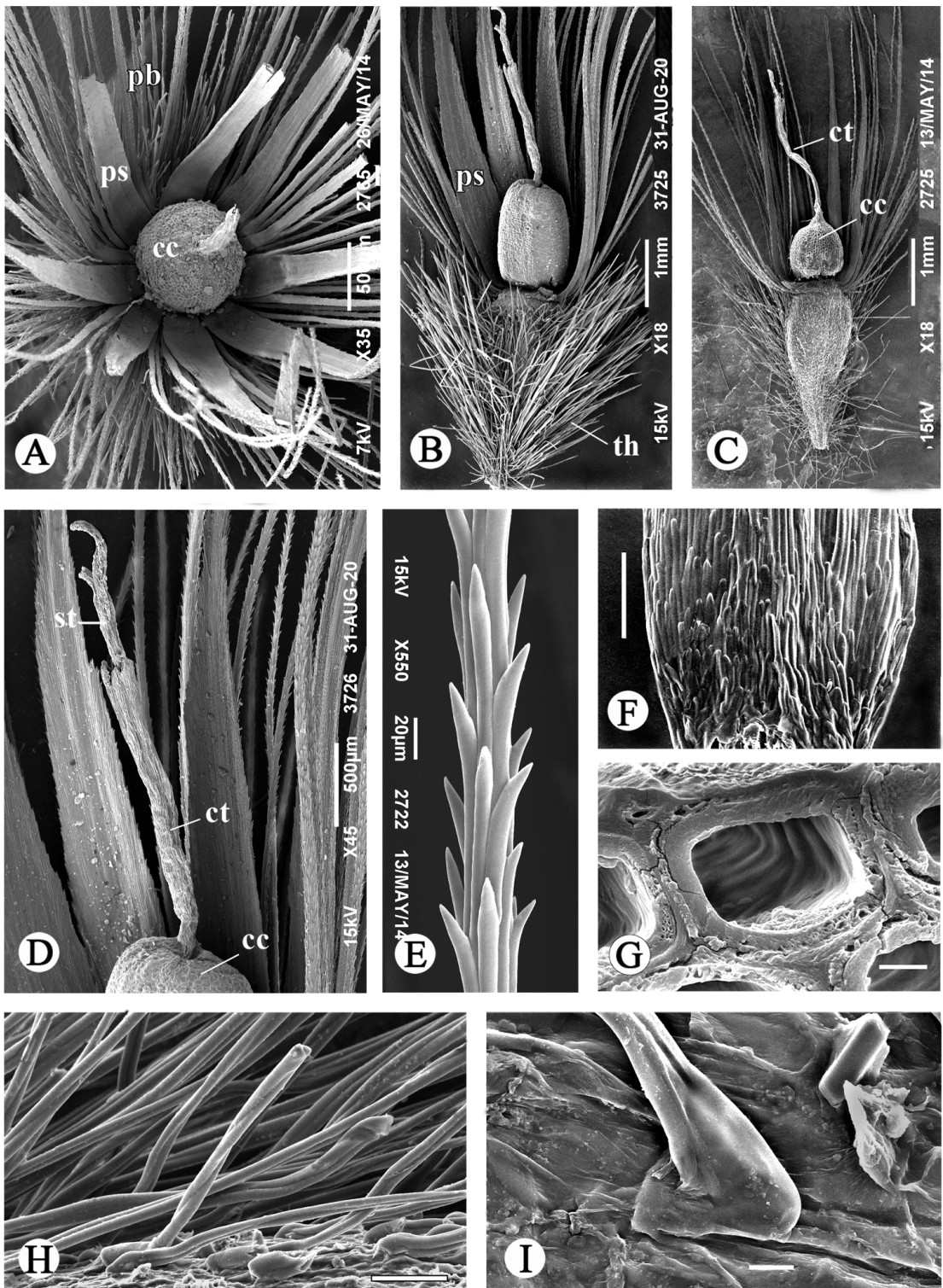

**Figure 3.** SEM images of aerial achenes and their structure in *G. micrantha*. (**A**) Achene with opened pappus, view from above. (**B,C**) Achenes with different sizes of the corolla cupula; the pappus elements and twin hairs partially deleted. (**D**) Apical achene part. (**E**) Fragment of pappus bristle. (**F**) Adaxial side of pappus scale. (**G**) Transverse section of the epidermal cell on the adaxial side of pappus scale. (**H**) Twin hairs of the pericarp. (**I**) Thickened base of twin hair. Abbreviations: cc, corolla cupula; ct, corolla tube; pb, pappus bristle; ps, pappus scale; st, style; th, twin hair. Scale bars: 1000 µm for (**B,C**); 500 µm for (**A,D**); 100 µm for (**F**); 50 µm for (**H**); 20 µm for (**E**); 10 µm for (**I**); 2 µm for (**G**).

### 3.1.2. The Subterranean Achenes

The achenes of type I are 3.0–5.2 mm long (including an underdeveloped pappus), obovate, slightly conical, not flattened or slightly flattened flat-convex, angular (Figures 1H and 4A,E(a,b)), brown or greenish, with a weak pubescence of sparse and short hairs mainly at the top of the achene. The primary sculpture of the achene surface is densely wrinkled or ruminated (Figure 4B). The corolla tube does not form a cupule. The corolla style and style branches are long, especially in small achenes (Figure 4E–G). The pappus consists of a few short, narrow, pointed at the end, thin scales and shorter bristles located almost in one row. There is a significant variation in the achene size within the partial fruiting head (Figure 4E(a,b)): the achenes adjacent to the large ones are almost two times smaller and contain medium-sized embryo (Figures 1J and 4E(b)). The embryos (Figure 1I,J), closely adjacent to the fruit wall, have massive cotyledons, significantly exceeding the hypocotyl-root axis in length.

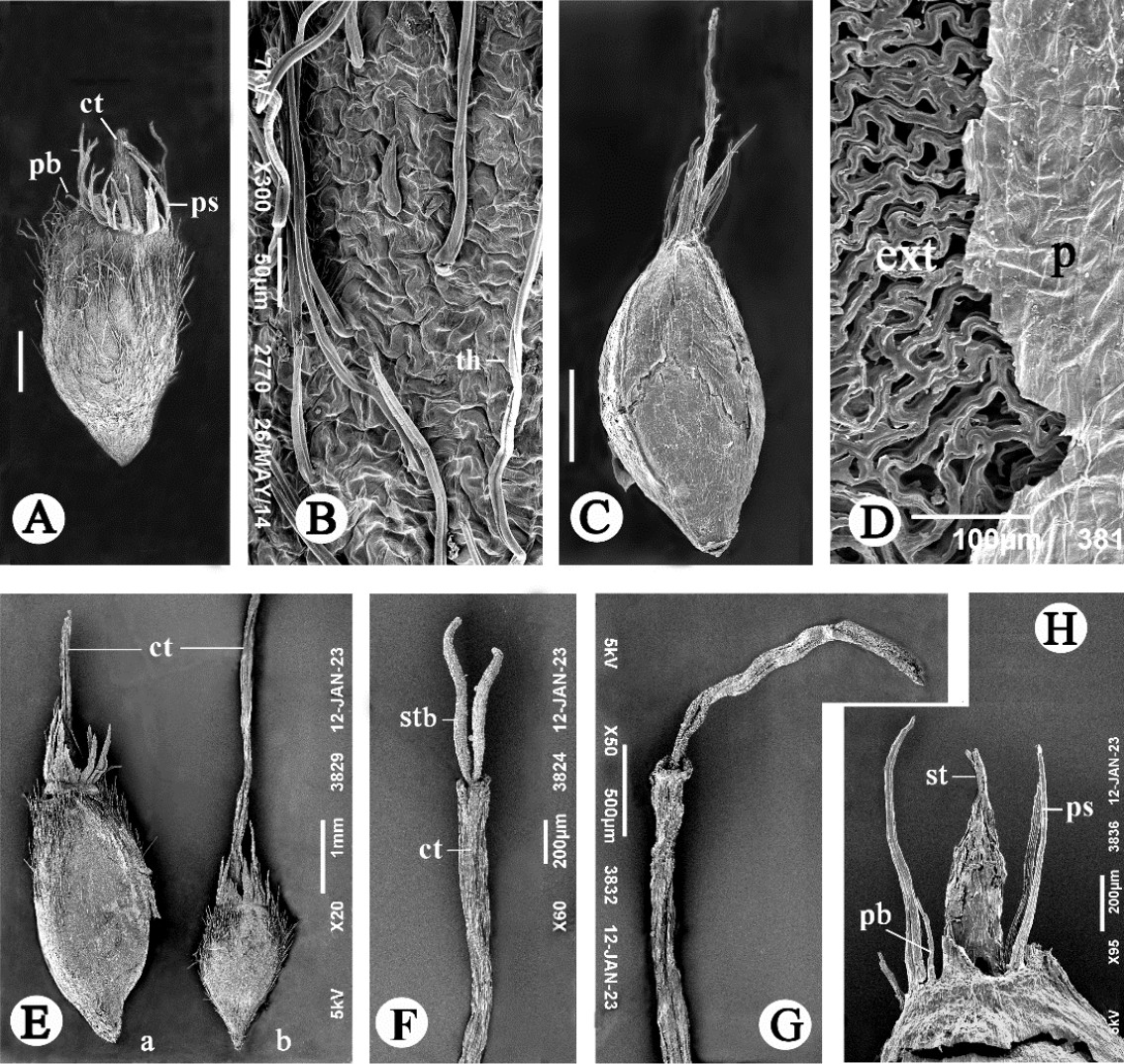

**Figure 4.** SEM images of subterranean achenes in *G. micrantha*. (**A**,**B**) Subterranean achene I and its surface, correspondingly. (**C**,**D**) Subterranean achene II and its surface, correspondingly. (**E**) Large (a) and small (b) subterranean achenes I from same two-seeded partial fruiting head. (**F**,**G**) Distal part of corolla tube with a protruding style in large and small achenes from Figure 4E(a) and Figure 4E(b), respectively. (**H**) Dissected apical part of subterranean achene II. Abbreviations: ct, corolla tube; ext, exotesta; p, pericarp; pb, pappus bristle; ps, pappus scale; st, style; stb, style branches; th, twin hair. Scale bars: 1000 μm for (**C**,**E**), 500 μm for (**A**,**G**), 200 μm for (**F**,**H**), 100 μm for (**D**), 50 μm for (**B**).

The achenes of type II are 5.5–6.4 mm long, elliptic, more or less rounded in cross-section, brownish-gray, with sparse, short hairs and with slightly prominent veins of membranous pericarp, which easily peels off (Figure 4C,D); exotesta also peels off readily from the inner layers of the seed coat. The corolla tube is either long, ribbon-like (Figure 4C), with a darkened, apparently non-functional style, or short, with a strongly shortened style located inside (Figure 4H). The underdeveloped pappus consists of a few, small, subulate elements (scales and shorter bristles) arranged almost in one row. The embryos are the same in shape and size as in subterranean achenes I.

### 3.2. Anatomical Structure of the Aerial Achenes

### 3.2.1. The Apical Plate

At the apical end of the aerial achene, under the corolla cupula and nectary, there is a concave apical plate, star-shaped in outline (Figure 5A–C and Figure 6B), along the edge of which a poorly developed apical pericarp crown is sometimes noticeable. The plate is bounded by the pappus, which is obliquely attached directly to the pericarp, to the side of its apical ring, consisting of a thin-walled compressed parenchyma (shown by the arrows in Figure 5A,B); it provides mobility to the pappus. A total of 4–5 layers of thick-walled and radially stretched sclereids form the apical plate. In the outer layer, they are large, loosely arranged and separated by large intercellular spaces (Figure 5C). The protruding edge of the apical plate is the pericarp apical ring, formed by the angled sclereids of the apical plate, sclereids and lignified parenchyma cells of the pericarp, which is thicker here than beneath, and lignified. A narrow ring of small, elongated sclereids runs along the edge (Figure 5C).

### 3.2.2. The Pappus

The pappus scales and bristles are formed by longitudinally elongated, non-lignified cells (Figure 5D) of various and often irregular shapes on the transverse section (Figure 5E–G). They differ in the wall thickness and are located quite randomly. The scales are plane-convex in section, with flat adaxial side. In the middle of the scale length, its membranous wing is 2-layered, formed by epidermal layers (Figure 5F); upper epidermis and subepidermis in the center of adaxial side are built of thick-walled cells. At the scale base, all epidermis in the adaxial scale side consists of such cells (Figure 5G,H). In surface view (Figure 3F), they are short and end up with small prominent apical teeth; their wall is thick and spongy, with transverse striation in the inner layer (Figure 3G). Pappus bristles vary from 3- to multicellular, gradually thinning to the outer row, and are more or less rounded, rounded-polygonal or irregular in transverse section (Figure 5E,G).

### 3.2.3. The Persistent Corolla and the Style

The corolla cupula is rounded, with parenchymatous wall (Figure 5E), which is thick, 6–7-layered at the base, and membranous at the top. It is weakly lignified or without lignin, and its cells have weak scalariform cell wall thickenings. The corolla cupula is close to the nectary (Figure 6E), and it is connected to the apical plate only by 4–6 or 9 corolla vascular bundles.

The style enclosed in the corolla cupula has a bulbous swelling 215–300 μm in diameter above the base (Figure 6A–F), and larger in larger cupules. The swelling is formed as a result of enlargement and radial proliferation of parenchyma cells in 1–2 subepidermal layers. The cell wall of most of these cells is thickened and provided with weak scalariform pits, gradually thinning to the periphery of the style. The epidermis consists of predominant large thin-walled cells with a convex outer periclinal wall, destroyed in some cells (Figure 6B,C), and rare small cells with a thickened wall, the outer periclinal wall being concave.

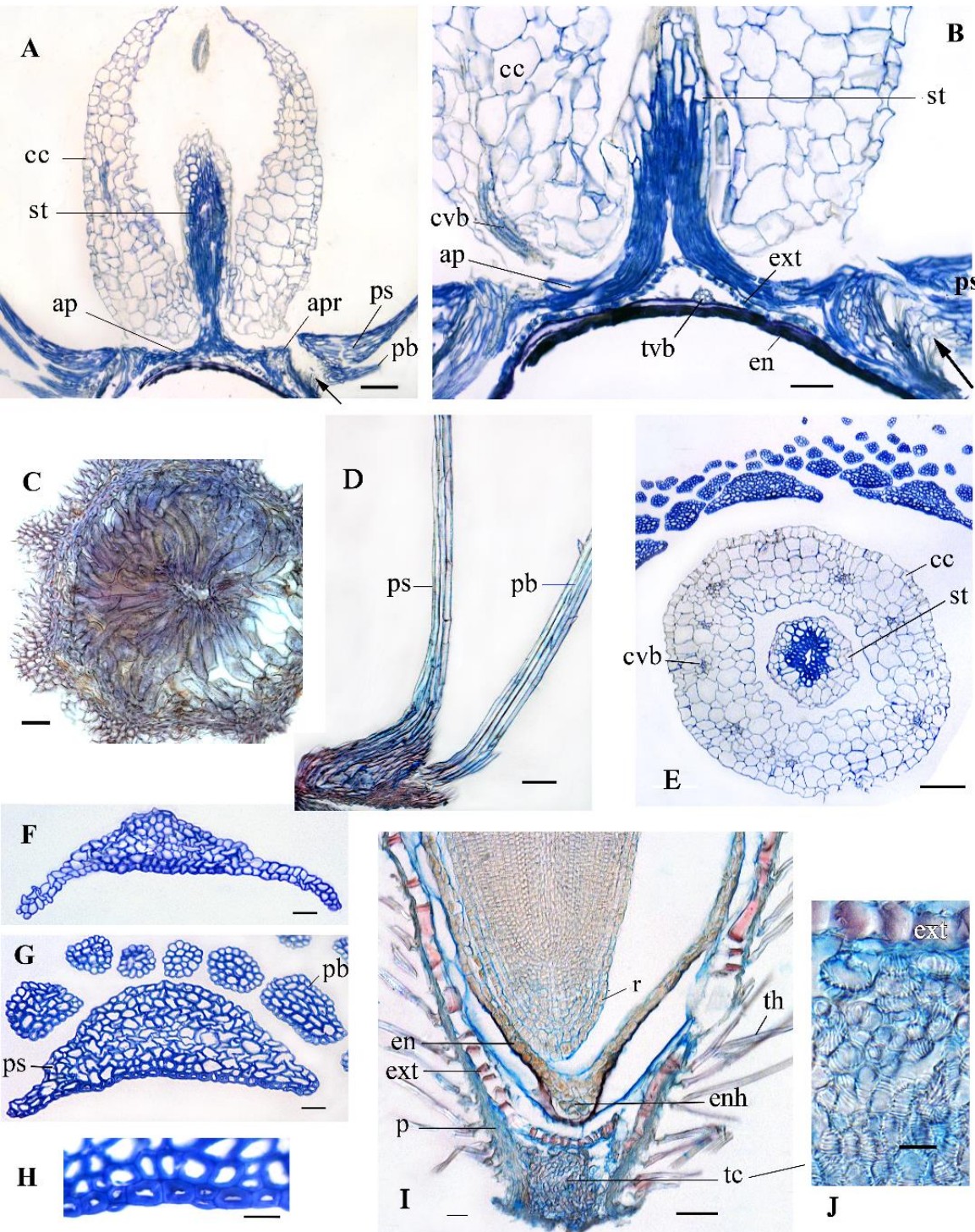

**Figure 5.** Anatomical structure of aerial achene in *G. micrantha*, light microscopy. (**A**,**B**) Longitudinal section (LS) of apical achene region at various magnifications. (**C**) Apical plate in transverse section (TS). (**D**) LS, pappus scale and bristle. (**E**) TS, corolla cupula and pappus base. (**F**) TS, the middle of pappus scale. (**G**) TS, the base of pappus scale and bristles. (**H**) TS, the adaxial side of pappus scale. (**I**) LS, basal achene region. (**J**) LS, tracheid-like cells in fruit stipe. Abbreviations: ap, apical plate; apr, apical pericarpal rim; cc, corolla cupula; cvb; corolla vascular bundle; en, endosperm; enh, micropylar endosperm haustorium; ext exotesta; p, pericarp; pb, pappus bristle; ps, pappus scale; r, radicle; st, style; tc, tracheid-like cells; th, twin hair; tvb, testa vascular bundle. Scale bars: 100 μm for (**A**,**E**), 50 μm for (**B–D**,**I**), 20 μm for (**F**,**G**), 10 μm for (**H**,**J**).

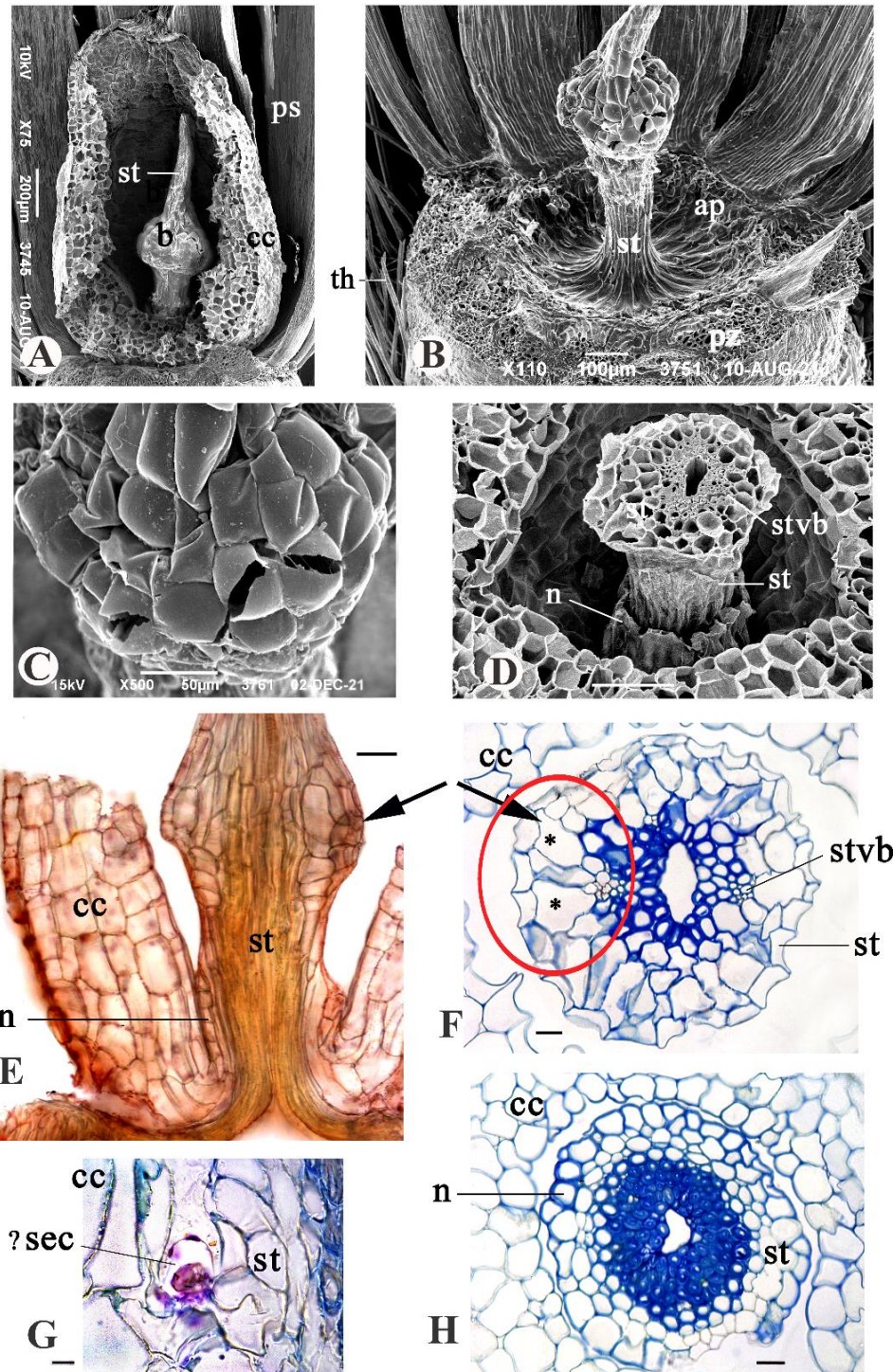

**Figure 6.** Apical region of aerial achene in *G. micrantha*, SEM (**A**–**D**) and light microscopy (**E**–**H**) images. (**A**) Dissected corolla cupula with a persistent style inside. (**B**) Apical plate and bottom part of the style; corolla, nectary and partly pappus deleted. (**C**) Surface of bulbous swelling of style. (**D**) Transverse section (TS) through the middle of corolla cupula and thickened style portion. (**E**) Fragment of longitudinal section (LS) through apical achene region. (**F**) TS, style thickened part. (**G**) LS, idioblast in the corolla axis. (**H**) TS, style base with nectary. Abbreviations: ap, apical plate; b, bulbous style swelling; cc, corolla cupula; cvb, corolla vascular bundle; n, nectary; pc, parenchyma cell; pz, pappus insertion zone; st, style; stvb, style vascular bundle; th, twin hair; ✱ large parenchyma cells. Scale bars: 200 μm for (**A**), 100 μm for (**B**,**D**), 50 μm for (**C**,**E**,**H**), 20 μm for (**F**), 10 μm for (**G**).

In the bulbous thickening of the style, the phloem of the vascular bundles borders on large subepidermal parenchyma cells, which are highly vacuolated (Figure 6E,F); there are wall protuberances and ruptures on the outer periclinal cell wall in certain adjacent epidermis cells. It looks like these structures (Figure 6E,F, shown by arrows and a red frame) serve for the accumulation and excretion of a secret; the appearance of breaks is facilitated by the tension that emerges in the parenchyma and the epidermis of the swelling due to the alternation of thin- and thick-walled cellular portions.

The stylar cup-shaped nectary with a finely lobed edge (sometimes only slight swelling) persists at the style base of *Gymnarrhena* aerial achenes (Figure 6D,E,H). It is thin, 1–2 layered, up to 6 cells in height, epidermal epithelium (term of I. Roth [8]), formed by cells with a thick lignified cell wall. The nectary separates by a gap from adjacent layers of thin-walled parenchyma. Moreover, there are idioblasts, possibly secretory cells, in the axil of the corolla tube (Figure 6G). The metachromasia is observed, when staining these cells with toluidine blue that indicates the presence of heteropolysaccharides in the secret.

### 3.2.4. The Basal Achene Region

The stipe of the aerial achene is short, loose with aerial cavities, provided with a plug of tracheid-like cells (Figure 5I,J). The seed vascular bundle is bent in the stipe. The micropylar haustorium of the endosperm, the remnants of the nucellus and the obliterated inner layers of the integument are retained in the seed base (Figure 5I).

### 3.3. Anatomical Structure of Subterranean Fruits

The apical and basal regions of the subterranean achenes have a simplified structure compared to the aerial ones. In the apical region of the achene I (Figure 7A,B), the pericarp, pappus elements, and the wall of the corolla tube are thinner; the last is only 1–2 cells thick. The sclereids of the apical plate are shorter than fiber-like sclereids of the apical plate in aerial achene. The nectary is absent. There is no close proximity between the style base and the corolla tube. The basal region of the subterranean achenes is shortened; there is no loose stipe filled with tracheid-like cells, but these cells were sometimes observed in the seed coat above the seed base. The micropylar endosperm haustorium consists of few cells (Figure 7D).

The involucres enveloping the subterranean achene morphs vary in their structure. The woody and colorless involucre of type I (Figure 7C) is 180–690 μm thick, almost completely lignified (except for a narrow zone of compressed thin-walled parenchyma under the subepidermis). The bracts forming it are greatly curved and folded, arranged in one or more rows. In their mesophyll, outside the inner fibrous zone 3–9 cells wide, there is a wide zone of large-celled lignified parenchyma (Figure 7C). Presumably single-leaf, coriaceous, dark brown involucre of type II (Figure 7E) is thinner, 295–440 μm thick. The parenchyma, located outward from the inner fiber zone, is unlignified, and contains tannins in cell walls.

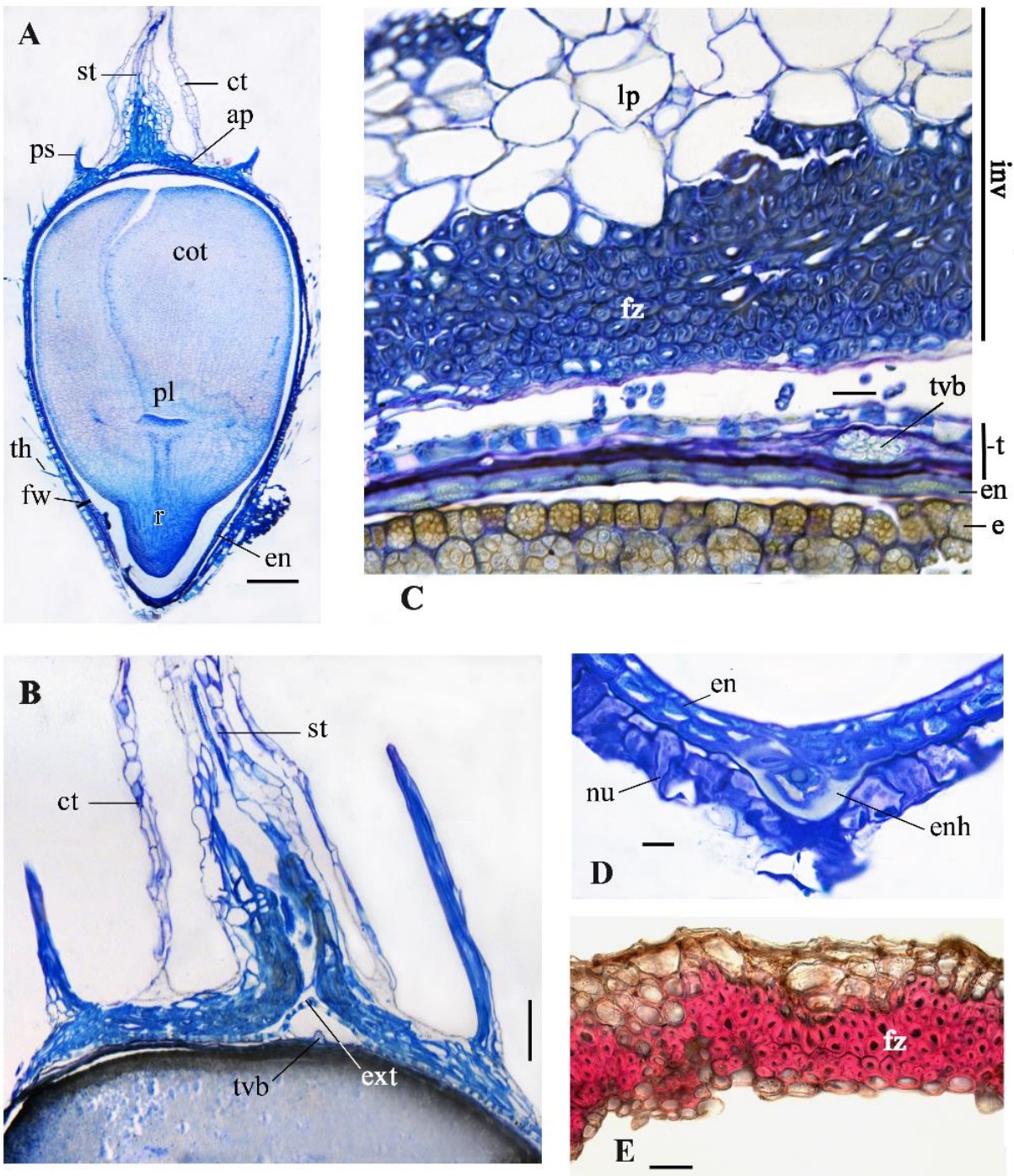

**Figure 7.** Anatomical structure of subterranean achenes in *G. micrantha*, light microscopy. (**A**) Longitudinal median section (LMS) of subterranean achene I. (**B**) LMS, apical achene region. (**C**) Fragment of transverse section (TS) through achene middle and adjacent involucral bract. (**D**) LMS, micropylar endosperm haustorium of subterranean achene I. (**E**) TS, involucre of subterranean achene II, the lignified tissue is colored red after treatment with phloroglucinol and sulfuric acid. Abbreviations: ap, apical plate; cot, cotyledons; ct, corolla tube; e, embryo; en, endosperm; enh, endosperm haustotium; ext, exotesta; fz, fiber zone; fw, fruit wall; inv, involucre; lp, lignified parenchyma; nu, nucellus remnants; pl, plumule; ps, pappus scale; r, radicle; st, style; t, testa (=seed coat); th, twin hair; tvb, testa vascular bundle. Scale bars: 200 μm for (**A**); 100 μm for (**E**); 50 μm for (**B**); 20 μm for (**C**); 10 μm for (**D**).

### 3.4. Fruit Wall Structure

The pericarp, seed coat and endosperm have a similar structure in aerial and subterranean achenes (Figures 7C and 8A–F), with the exception of a dense pubescence of the pericarp, which is present only in aerial achenes.

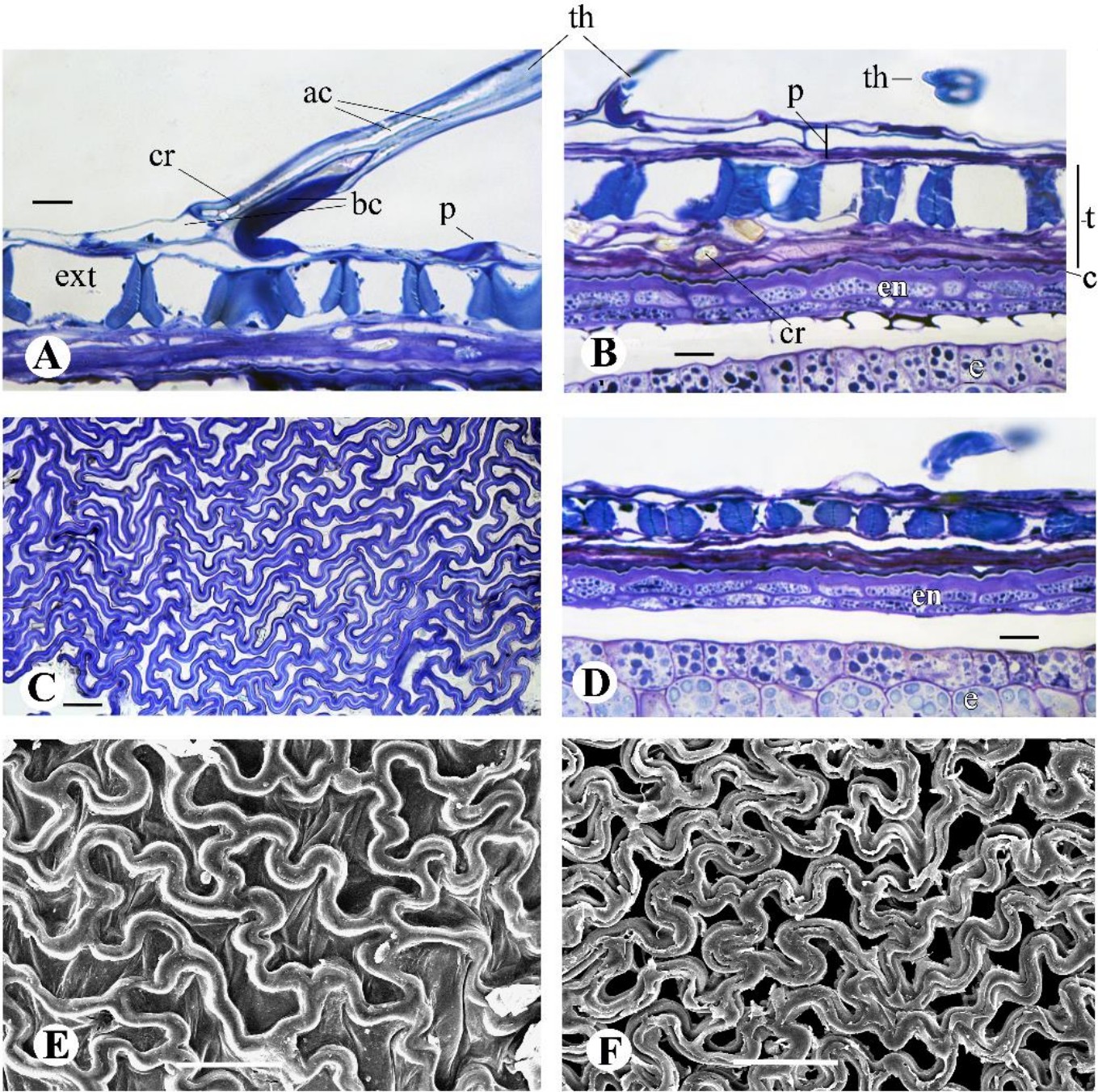

**Figure 8.** Fruit wall structure in *G. micrantha*, light (**A–D**) and scanning electron (**E,F**) microscopy. (**A,B**) Longitudinal sections (LS) of the achene wall in the basal region of aerial achene (**A**) and subterranean achene I (**B**). (**C**) Exotesta in the middle of subterranean achene I in surface view (SW). (**D**) LS, the middle of subterranean achene I. (**E,F**) SW, SEM micrographs of exotesta in subterranean achenes: (**E**) achene I, (**F**) probably subterranean achene II. Abbreviations: ac, apical cells of twin hair; bc, basal cells of twin hair; c, cuticle; cr, Ca oxalate crystal; e, embryo; en, endosperm; ext, exotesta; p, pericarp; t, testa (=seed coat); th, twin hair. Scale bars: 50 μm for (**C,E,F**); 10 μm for (**A,B,D**).

### 3.4.1. The Pericarp

It is very thin, membranous, poorly visible and formed on most of the fruits by two layers of strongly compressed thin-walled parenchyma cells without content (Figure 8A,B), at the base of the aerial achenes by 2–3 or more layers. Exocarp cells are polygonal, covered with a thin cuticle. There are usually four vascular bundles in the pericarp, which is typical for Asteraceae [11], but in the largest aerial achenes there are six. Pericarp hairs of the aerial achenes 500–2000 μm in length are typical twin hairs [45], with a hygrophilous cell at the base (Figure 8A). In the lower part of the outer apical cell, there are crystals, sometimes large (Figure 3I). Apical cells are mostly of the same length, shortly forked at the apex. Basal cells differ significantly from each other, the inner one is a specialized hygrophilous cell, and the outer thin-walled cell is similar to ordinary epidermal cells. Other hair cells have a thickened cell wall. Pericarp twin hairs in subterranean achenes are sparse, especially in achenes II. They are much smaller than twin hairs of aerial achenes (Table 1) and have a simplified structure (Figure 8B). The hygrophilous basal cell is undeveloped, which may indicate that the hairs do not have the function of absorbing and retaining water.

**Table 1.** Fruit wall structure of heteromorphic achenes in *Gymnarrhena micrantha*. "-": absence of a character; "±": weak development of a character; measurements are given in μm.

| Character | Aerial Achene | Subterranean Achene I | (?)Subterranean Achene II |
|---|---|---|---|
| **Delamination of the fruit wall** | - | - | + |
| **Pericarp hairs** | | | |
| Density | densely arranged | rare | very rare |
| Length | 500–2000 | <500 | <500 |
| **Seed coat in the middle of the seed** | | | |
| Length of exotesta cells | 70–135 | 110–170 | 102–170 |
| Thickness of anticlinal cell walls in exotesta | 8–12 | 9–18 | 12–18 |
| Orientation of crystals along the seed | ± | - | - |
| **Endosperm in the middle of the seed** | | | |
| Cells length in the outer layer | 20–40 | 25–50 | 27–60 |
| Thickness of anticlinal cell walls in the outer layer | 1.5–6.0 | 1.5–4.0 (5.0) | 1.5–3.0 |

### 3.4.2. The Seed Coat (Testa)

The seed coat consists of an exotesta covered with a thin cuticle and compressed inner layers of thin-walled cells. It is 2–3 times thicker in the lower part of the achene (Figure 8A,B) than in the middle and upper parts (Figure 8D) due to larger exotesta cells and better preservation of its inner layers. In both aerial and subterranean achenes, its thickness varies from 15–20 μm at the apical end to 27–35(45) μm at the base. Cells of inner layers contain a large number of Ca oxalate crystals that are mostly narrowly 6-angled, lamellar; they are oriented mainly along the seed in the aerial achenes; in the subterranean ones, their orientation is absent. The entire fruit wall at the achene base is thicker also due to larger endosperm cells and the presence of nucellus remnants, in the aerial achenes, in addition, due to a thicker pericarp.

The exotesta cells, oriented mainly transversely and obliquely, are flattened and irregularly shaped with curved and sinuous anticlinal walls (Figure 8C,E,F). Their outer and inner periclinal walls are thin, the anticlinal walls are thickened and lignified (collenchyma according to Dittrich [42]), 8–18 μm thick, thicker in the subterranean achenes than in aerial ones (Table 1). This exotesta structure was described as type *Gongylolepis* in the tribe Mutisieae, Asteraceae [46] and type *Jurinea* in Carduinae, tribe Cardueae [43]. In different parts of *Gymnarrhena* achenes, exotesta cells differ in size, shape and shape of cell wall thickenings. In the middle and apical regions, they have a slit-like cavity and form an almost continuous lignified seed cover (Figure 8C); their anticlinal walls are mostly cordate in section in aerial seeds and cordate, rounded or elliptic in subterranean seeds (Figure 8D). In the lower seed part (Figure 8A,B), the exotesta cells are larger, their anticlinal walls are

sagittate (aerial seeds) or sagittate and trapezoid (subterranean seeds) in section. The upper edge of the anticlinal cell walls of the exostesta observed by SEM after separation of the pericarp (Figure 8E,F) is different in subterranean achenes of types I and II: in type I it is smooth (Figure 8G), in type II it is rough and grooved, with remnants of the outer periclinal cell wall (Figure 8F).

### 3.4.3. The Endosperm

The endosperm is 2-layered in most of the seed (Figures 7C,D and 8B,D), 2–3-layered at the level of the plumula, covered with a cuticle and not separated from the testa. It is thinner (6–10 μm) at the seed top and in the middle than in other seed parts (up to 25 μm). Its cells are thick-walled, polygonal, flattened, less flattened in the outer layer, and filled with proteins and oils. In the outer layer, the outer periclinal cell walls are strongly thickened and have a wavy outer border on the section. The cell length and the thickness of their anticlinal walls differ between morphs (Table 1): in the aerial seeds, the endosperm cells are smaller, but thicker walled compared to the subterranean seeds. The micropylar haustorium of the endosperm, consisting of large living cells with a strongly thickened, lipid-containing outer periclinal wall, is smaller in subterranean seeds than in aerial ones (Figures 5I and 7D).

## 4. Discussion

### 4.1. Fruit Diversity in Gymnarrhena

The present study showed that more fruit types occur in *G. micrantha* than it was previously known, of which there are five (Figure 1C–E,H,L). Two additional aerial morphs arise due to the persistent, basally attached bract twisting tightly around one or two achenes. Their appearance is the result of hygroscopic movements of several organs and various tissues [20]. This process involves both subtending bracts and the aerial achenes themselves, which actively repel the bracts with the opening pappus and protruding hairs. Such behavior of the receptacular bracts suggests its significant morphogenetic potential. Stuessy and Spooner [47] attached great adaptive importance to receptacular bracts and discussed their functions in Asteraceae; one of the principal functions is dispersal and protection. In *G. micrantha*, the receptacular bracts, subtending each achene in the fruiting aerial head, are tightly closed, and protect achenes from maturation in spring to their dispersal in winter rainy season. After getting wet and re-drying, their function is different: attached to the pappose achenes they do not aid to their flight, but slow it down, thus providing, obviously, the differentiation of wind-dispersed fruits in flight range. In addition, in arid climates, receptacular bracts that attached the achene may keep moisture around it [47].

The aerial achenes of *G. micrantha* also differ within the same plant in the size of the apical cupula, which affects their weight. The size of the corolla's cupula is not, as we assume, an indicator of the fruit maturity. Further detailed field observations are required to investigate this issue, as well as to determine to what extent the increasing of aerial achenes weight at the expense of large cupula or bract affects their ability to fly.

In subterranean fruiting heads, the involucral bracts perform the protective function. They closely envelope the epappose achenes, forming involucrate fruit—a new dispersal unit of different morphological nature, with hard accessory cover. The involucres surrounding subterranean achenes of two types vary in color, consistency and anatomical structure.

The results obtained show that *G. micrantha* is characterized by divergent differentiation of diaspores, an increase in their diversity along several lines of morphological differentiation which corresponds to the multiple strategies of seed dispersal and germination. The plurality of reproductive strategies and different dissemination modes apparently allow this species to adapt to a wide range of habitats, which distinguishes it from the majority of other winter annuals in desert region [20]. Imbert [10] also emphasized that for the heterocarpy in Asteraceae the involvement of several characters in fruit differentiation is more typical than in other families.

*4.2. From Which Kind of Flowers Do Subterranean Fruits Develop?*

The data on the nature of *Gymnarrhena* flowers, from which subterranean fruits develop, are still contradictory. Many authors incorrectly note the presence of cleistogamous flowers in *Gymnarrhena*, while the others consider them chasmogamous cross-pollinated ground-level flowers that are pulling into the soil shortly after pollination by insects [4,23,32] based on Plitmann's observations of Israel flora. This is consistent with Koller and Roth's [20] observation that the corolla tubes of subterranean flowers are open at the surface when flowering. Zohari [1] also considered these flowers chasmogamous, but subterranean ones; he assumed that their pollination occurs within the subterranean heads (geitonogamy) that include closely spaced pistillate and staminate flowers. Furthermore, as in the case of *Emex spinosa* [23], cleistogamous flowers, in which self-pollination occurs, are theoretically impossible in *Gymnarrhena*, because the fruit develops from an unisexual pistillate flower. Zohary [31] (p. 96), back in 1930 said, "the amphicarpy in *G. micrantha* deserves special interest, because this phenomenon in plants with unisexual flowers is still unknown or little is known".

Our results confirmed that subterranean achenes I could develop from chasmogamous flowers in *G. micrantha*. They have a long persistent style and corolla tube, especially long in small achenes, apparently reaching the top of the subterranean head. In addition, following Zohary [1,31], we found staminate flowers within the subterranean heads (at least in some ones).

Subterranean achenes II, enclosed in a bud-like involucre, have an underdeveloped style, which, apparently, does not function. It suggests that the embryo arises, most likely, through apomixis. Thus, a mixed mating system that includes potentially outcrossing, apomixis and, perhaps, autogamy (geitonogamy) can be assumed for this species.

*4.3. Structural Differences of Heteromorphic Fruits of Gymnarrhena*

The study showed that the subterranean achenes of *G. micrantha*, in addition to the number of already known features (larger size, underdeveloped pappus, weak pubescence, accessory envelope of bracts), have some additional differences from the aerial achenes. These include: the simplified structure of the apical and basal regions, the absence of corolla cupula, the nectary, the loose stipe filled with tracheid-like cells, somewhat thicker cell walls of the exotesta, larger and thinner-walled endosperm cells, different embryo shape (with disproportionately massive cotyledons).

Probably subterranean achenes II are morphologically similar to achenes I. Their distinctive features are a slightly larger size, a greyish color, with the pappus formed mainly by styloid elements, even sparser pubescence, and the ability of the fruit wall to exfoliate with separation of the pericarp and exotesta from the internal seed coat layers, thicker and heterogeneous exotesta cell walls and larger thinner-walled endosperm cells. A characteristic feature of achenes II is their rapid drying when removed from the involucre. In the involucre, unlike subterranean type I of the fruit, mechanical tissue is less developed, and there is no wide outer zone of large-celled air-containing lignified parenchyma capable of storing water. The differences between the seeds in two subterranean morphs in germination behavior can be assumed since it is known that the morphology of the surrounding bracts affects the dormancy of heteromorphic fruits [10].

An unusual feature of *Gymnarrhena* subterranean achenes I is that they germinate faster than aerial achenes [20], although they are enclosed in a massive additional lignified involucre. Persistent involucral bracts enveloping the fruit usually create a barrier to water absorption that delays germination [10,48]. Imbert [10] quotes Gutterman's statement that their adaptive significance lies in the fact that the involucre prevents premature germination; thus, the germination of achenes of some desert Asteraceae species is only possible with heavy rainfall, which increases the likelihood of seedling survival. In *G. micrantha*, the shallow dormancy of underground achenes may be because they are already sown in the soil [9]. We assume that several structural features may be related with their rapid germination: the absence of the loose stipe (in the presence of water, it penetrates faster to

the radicle), less compacted endosperm tissue than in aerial achenes [19] and the absence of dense cover of hairs.

In amphicarpic species, "subterranean fruits are often underdeveloped as compared with the aerial ones" [11] (p. 597). A striking example of this is the amphicarpic legume *Amphicarpaea edgeworthii*, in which subterranean fruits and seeds have a simplified structure of the pericarp and seed coat [16–18], and the seed coat signs associated with its water impermeability and hardness disappear. The main difference in the structure of the fruit wall between *Gymnarhena* aerial and subterranean achenes is that the pericarp of the last ones lacks numerous trichomes. In aerial hairy achenes, the twin hairs not only help to separate subtending bract, providing aid to anemochory and anchorage of the achenes to the substrate, but also absorb and retain water, and seemingly protect the fruit from drying out. Appressed to the pericarp surface and composed of thick-walled cells containing Ca oxalate crystals in the basal part, these trichomes form a protecting fruit envelope. Their absence in subterranean achenes probably makes the fruit wall more water permeable, which may affect the rate of germination. It can also be assumed that the spare nutrients that were not used for the formation of dense pubescence and pappus instead were utilized in the development of a large embryo.

The presence of both large and relatively small achenes of type I within one involucre may lead to differences in the time of their germination. The larger and disproportionately developed embryo (with greatly enlarged cotyledons) is obviously related with a larger and more developed robust seedling developing from subterranean seeds [20].

*4.4. Possible Adaptive Significance of Aerial Achene Structural Features*

In *G. micrantha*, the pappus is a multifunctional structure that performs all the functions noted for it in Asteraceae: it protects the head during anthesis and fruit ripening; separates mature achenes from neighboring fruits and subtending bract when wet; aids in the dispersal by wind (anemochory); and fixes the achenes on the soil [2,43,49,50]. Its "opening" when wet (hygrochasia) determines the time of the onset of dispersal. It has been noted that hygrochasia is a mechanism of dissemination, usually associated with arid conditions; dissemination is thus limited in time and can only occur when the water necessary for germination is available [39].

The results obtained show that the mechanism of hygroscopic (hygrochasic) movements of the pappus scales in *Gymnarrhena* corresponds to the Cirsium type [51], in which the movement is due to unequal contraction or swelling of the outer and inner sides of the pappus elements. In plants belonging to this type, the pappus is always well developed; the scales are characterized by a significant differentiation of cells into thin-walled parenchymatous and mechanical thick-walled cells, differing in their ability to swell. In *G. micrantha*, the epidermis and subepidermis on the adaxial scales side consist of thick-walled cells, which greatly increase in volume when wet; water absorption is facilitated by their spongy cell wall (Figure 3H). Neighboring thin-walled cells of inner scale layers remain unchanged, which creates tension and bending of the pappus scales and bristles.

For the first time, a specialized apical structure of aerial achenes of *G. micrantha* is described, which is an expanded base of the corolla tube. We call it corolla cupula and believe that it is an adaptation to dissemination by rainwater, ombrohydrochory [33]. This hollow, weakly lignified structure, with a hard lignified style in the center, can serve as a "float" that provides buoyancy to the achene. In addition, when disseminated by both water and wind, it gives the achene a vertical position in which the radicle is oriented downwards. The need for vertical orientation, apparently, is associated with the special structure of the fruit stipe, the plug, filled with loose tissue of tracheidal cells. These cells are believed to perform a water-conducting and water-storing functions [52,53]. In seeds, they may serve as a vascular structure complementary to the main vascular system and in Papilionoideae seeds, where they are situated near the hilum, they regulate water uptake and loss [54]. In *Gymnarrhena*, their presence can contribute similarly to the water rapid absorption and accumulation in the achene basal region, and control water entry into the

seed, which is obviously important for germination in arid habitats. When they are wet for a long time, the corolla cupula will fill with water (although the air always remains in its upper part) and can serve (as well as the style remaining inside it) as a reservoir for water storing.

The peculiar structure of the apical achene region, in our opinion, also indicates its possible function of attracting ants. Heavily wet and deformed achenes with adhering colorless, translucent and partially destroyed hairs and pappus (likewise, the dissected fruits in Figure 3B,C) are very reminiscent of termites due to the cupula; there is a similarity also in size and color with them. In addition, ants are known to be the main eaters of termites. In our opinion, *G. micrantha* demonstrates a case of mimicry in plants. The dissemination-promoting similarity of fruits and seeds with insects is known in representatives of many angiosperm families [2,55], including Asteraceae. However, in the most famous cases (seeds of *Melampyrum* and *Ricinus*, larva fruits of *Calendula*), this similarity, according to the above authors, is considered adaptive to attract or repel birds.

Some of our observations made on herbarium material and concerning secretory structures in mature achenes are difficult to interpret unambiguously. In particular, it can be assumed that the structures located in the thickened part of the style opposite the vascular bundles (shown by arrows and the red frame in Figure 6E,F) serve to accumulate and excrete a secret through ruptures in the outer wall of epidermal cells. The corolla cupula may be filled with an ant-attracting nectar secreted by several nectaries within it: besides typical for Asteraceae tubular nectary [56,57], additional secretory structures were found in the corolla axil. Ants have a uniquely fine olfaction, which is their most developed sense [55]. Ant-attracting pericarpial post-floral nectaries formed after anthesis from floral nectaries, which continue to secrete a nectar, are known for many Rubiaceae species [58,59]. Van der Pijl [2] also suggests that some diaspores, such as those of *Melampyrum pratense*, attract ants by postfloral nectar.

## 5. Conclusions

The results obtained revealed extraordinary abilities in *Gymnarrhena micrantha* to increase the structural fruits diversity. This species is characterized by the divergent differentiation of diaspores and their transformation along several lines of morphological differentiation, which correspond to the multiple strategies of seed dispersal and germination. The number of diaspores found (five) is the same as that of another amphicarpic composite species *Catananche lutea*; both are annuals of dry habitats. The results of this work suggest that *Gymnarrhena* has a mixed mating system, which includes outcrossing, apomixis and, perhaps, autogamy (geitonogamy).

Some additional differences were clarified between heteromorphic achenes of *G. micrantha*. The subterranean epappose achenes are characterized by a simplified structure of the apical and basal regions and a large embryo with disproportionately developed and massive cotyledons. The absence of any external achene structures was noted in subterranean achenes that may compete with the embryo for water. Moreover, we hypothesize that the disappearance of dense pericarp pubescence could lead to an increase in the water permeability of the subterranean achene wall and affect germination.

Of particular interest is the question of the adaptive significance of the specialized apical structure of aerial achenes in *G. micrantha*, formed by the swelling base of the corolla tube. Perhaps, corolla cupula is an adaptation to dissemination by rainwater and can serve as a "float" that provides the achene with buoyancy and vertical orientation, and when wet for a long time, as a reservoir for storing water. The shape acquired by the aerial achene due to the apical cupula and the presence of additional secretory structures inside the cupula also suggest another function associated with attraction of ants (synzoochory).

In many respects, *G. micrantha* remains an incompletely studied plant. Further research of this species and field observations are required to clarify some issues of its reproductive biology, in light of the results obtained. The hypotheses put forward in this work need to be confirmed by experimental data.

**Funding:** This research was funded by institutional research project no. AAAA-A18-180316900084-9 of the Komarov Botanical Institute of Russian Academy of Sciences. The author participated in this study in a project supported by the Russian Foundation for Basic Research under grant 13-04-0852.

**Institutional Review Board Statement:** Not applicable.

**Informed Consent Statement:** Not applicable.

**Data Availability Statement:** Not applicable.

**Acknowledgments:** The work on *Gymnarrhena* carpology was started by Marianna Arsenyevna Plisko (1941–2018), and I thank her for some compiled materials and some drawings. The author appreciates the help of the curators of Herbarium LE, from which the material was received, and is grateful to L.A. Kartseva for her technical assistance in SEM studies, A. Moore who provided the literature, N.K. Koteeva, L.E. Muravnik, O.V. Yakovleva and G.Yu. Konechnaya for valuable advice when writing this article, J.V. Osadtchiy and A.N. Ivanova for their help with translation into English, and to the staff of the Laboratory of Plant Anatomy and Morphology for preparing some material for investigation and assistance in photography.

**Conflicts of Interest:** The author declares no conflict of interest.

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
