# Peer review of "Fruit Structure in Amphicarpic Annual Gymnarrhena micrantha (Asteraceae, Gymnarrheneae) in Relation to the Species Biology"

_2037-0164, doi:10.3390/ijpb14020032_

Round 1
Reviewer 1 Report
This is a good descriptive study on fruits of the highly heterocarpic Gymnarhena micrantha, which provides insights into anatomical structures of its fruit variants and their possible significance. A minor drawback of this study is its being based on old herbarium specimens rather than on fresh field collections and observations; nevertheless, the study is still rich in results and conclusions.
This is a nice anatomical study, useful and worth of publishing. I have no improvement to the contents.
A weak point of this submission is its English, which is moderately flawed. I suggested some improvements in the uploaded file (annotated manuscript). After these changes, the whole manuscript should be linguistically proofread once again.

Reviewer 2 Report
This is an excellent piece of research. Well presented with exquisite illustrations. The results are very interesting and will contribute significantly to the reproductive biology of the species Furthermore, it adds to our understanding of the Asteraceae as a whole and provides the basis for the understanding of this fruit type in the family.
The manuscript could use some copy editing for English usage but that is minor and easily accomplished at the proof stage.
This is a fine piece of research in integrating several anatomy approaches with ecology and reproductive biology. I am pleased to see this coming from the Komarov Botanical Institute. Armen Tahktajan would have enjoyed reading this as much as I have.
Dennis Wm. Stevenson
